# Targeting host deoxycytidine kinase mitigates *Staphylococcus aureus* abscess formation

**Volker Winstel[1,2]\*, Evan R Abt[3], Thuc M Le[3], Caius G Radu[3]**

[1]Research Group Pathogenesis of Bacterial Infections; TWINCORE, Centre for Experimental and Clinical Infection Research, a joint venture between the Hannover Medical School and the Helmholtz Centre for Infection Research, Hannover, Germany; [2]Institute of Medical Microbiology and Hospital Epidemiology, Hannover Medical School, Hannover, Germany; [3]Department of Molecular and Medical Pharmacology, David Geffen School of Medicine, UCLA, Los Angeles, United States

**\*For correspondence:** winstel.volker@mh-hannover.de

**Sent for Review** 06 August 2023
**Preprint posted** 18 August 2023
**Reviewed preprint posted** 11 October 2023
**Reviewed preprint revised** 05 March 2024
**Version of Record published** 21 March 2024

**Abstract** Host-directed therapy (HDT) is an emerging approach to overcome antimicrobial resistance in pathogenic microorganisms. Specifically, HDT targets host-encoded factors required for pathogen replication and survival without interfering with microbial growth or metabolism, thereby eliminating the risk of resistance development. By applying HDT and a drug repurposing approach, we demonstrate that (*R*)-DI-87, a clinical-stage anticancer drug and potent inhibitor of mammalian deoxycytidine kinase (dCK), mitigates *Staphylococcus aureus* abscess formation in organ tissues upon invasive bloodstream infection. Mechanistically, (*R*)-DI-87 shields phagocytes from staphylococcal death-effector deoxyribonucleosides that target dCK and the mammalian purine salvage pathway-apoptosis axis. In this manner, (*R*)-DI-87-mediated protection of immune cells amplifies macrophage infiltration into deep-seated abscesses, a phenomenon coupled with enhanced pathogen control, ameliorated immunopathology, and reduced disease severity. Thus, pharmaceutical blockade of dCK represents an advanced anti-infective intervention strategy against which staphylococci cannot develop resistance and may help to fight fatal infectious diseases in hospitalized patients.

## eLife assessment

This **valuable** study combines *in vitro* and *in vivo* experiments designed to test if a deoxycytidine kinase inhibitor provides therapeutic benefit during infection with *Staphylococcus aureus*. The authors provide **compelling** evidence that this putative host-directed therapy has good potential to promote natural clearance of infection without targeting the bacterium. This paper would be of interest to bacteriologists, immunologists, and those studying host-microbe interactions.

## Introduction

Hospital- and community-acquired infections caused by antibiotic-resistant bacterial pathogens represent a global public health threat (*Wenzel, 2007*; *GBD 2019 Antimicrobial Resistance Collaborators, 2022a*; *GBD 2019 Antimicrobial Resistance Collaborators, 2022b*). Specifically, infections caused by multidrug-resistant microbes are often associated with increased morbidity and mortality, as compared to infections caused by antibiotic-sensitive strains (*Klevens et al., 2008*; *Yaw et al., 2014*; *GBD 2019 Antimicrobial Resistance Collaborators, 2022a*; *GBD 2019 Antimicrobial Resistance Collaborators, 2022b*). In addition, the pathology of various bacterial infections includes multifactorial

manipulation of host multicellular assemblies that enable pathogen replication, dissemination of disease, and adaptation to new hosts (*Foster, 2005*; *Finlay and McFadden, 2006*; *Okumura and Nizet, 2014*; *Thammavongsa et al., 2015*). Thus, prevention and therapy of such infections represent a major challenge in clinical practice and frequently require administering last-resort antibiotics such as vancomycin, linezolid, or colistin (*Chambers and Deleo, 2009*; *McKenna, 2013*; *Tong et al., 2015*). However, many clinically relevant pathogens have already developed resistance-mechanisms against many of these drugs thereby rendering antibiotic therapy of appropriate infections often ineffective (*Meka and Gold, 2004*; *Chambers and Deleo, 2009*; *GBD 2019 Antimicrobial Resistance Collaborators, 2022a*; *GBD 2019 Antimicrobial Resistance Collaborators, 2022a*). Moreover, the rapid exchange of antibiotic resistance genes via horizontal gene transfer events, along with the large-scale usage of antimicrobials in hospitals and in the animal industry, continuously triggers the emergence of new drug-resistant and globally spreading bacterial clones, highlighting the desperate need for novel treatment options to fight infectious diseases in human or animal hosts (*Chambers and Deleo, 2009*; *Nübel, 2016*; *Jian et al., 2021*).

Antibiotic-resistant bacteria are frequently associated with local and life-threatening infections in both, hospital environments and in the general community. Of particular concern are Mycobacteria, certain Gram-negative as well as Gram-positive bacteria including *Staphylococcus aureus*, a notorious pathogen that colonizes approximately 30% of the human population (*Lowy, 1998*; *Kuehnert et al., 2006*; *Lee et al., 2018*). *S. aureus* causes skin and soft tissue infections and is a widespread cause of endocarditis, septic arthritis, osteomyelitis, bacteremia, and sepsis (*Lowy, 1998*; *Fridkin et al., 2005*; *Klevens et al., 2007*). Moreover, this microbe triggers toxic shock syndrome, food poisoning, ventilator-associated pneumonia, and associated infections of the respiratory tract (*Lowy, 1998*; *Lee et al., 2018*; *Torres et al., 2021*). Nonetheless, a hallmark of staphylococcal disease is the formation of suppurative abscesses, where *S. aureus* provokes inflammatory responses that attract neutrophils, macrophages, and other immune cells to the site of the infection (*Lowy, 1998*; *Cheng et al., 2011*; *Spaan et al., 2013*). Abscess formation is further accompanied by liquefaction necrosis, the release of purulent exudates into circulating body fluids, and several defensive host responses designed to limit staphylococcal replication and spread (*Cheng et al., 2011*). For example, staphylococcal invasion of host tissues and abscess formation includes deposition of structural fibrin clusters, which shield healthy tissues from disseminating staphylococci (*Cheng et al., 2011*). Moreover, neutrophils combat *S. aureus* by powerful mechanisms which involve phagocytosis, generation of reactive oxygen species, and the formation of neutrophil extracellular traps (NETs) (*Spaan et al., 2013*). NETs represent an extracellular matrix composed of nuclear and mitochondrial DNA spiked with antimicrobial peptides, cell-specific proteases, and granular proteins that together ensnare and kill microbial invaders (*Brinkmann et al., 2004*). However, *S. aureus* rapidly evades NET-mediated killing due to the secretion of a potent nuclease (Nuc) capable of disrupting these structures (*Berends et al., 2010*; *von Köckritz-Blickwede and Winstel, 2022*). Intriguingly, Nuc-mediated degradation of NETs leads to the release of deoxyribonucleoside monophosphates that can be converted by the staphylococcal cell surface protein AdsA (adenosine synthase A) into death-effector deoxyribonucleosides (*Thammavongsa et al., 2013*; *Tantawy et al., 2022*). Notably, AdsA-derived death-effector deoxyribonucleosides such as deoxyadenosine (dAdo) and deoxyguanosine (dGuo) display toxigenic properties and contribute to immune cell death during acute and persistent *S. aureus* infections (*Thammavongsa et al., 2013*; *Winstel et al., 2019*; *Tantawy et al., 2022*). More precisely, dAdo and dGuo-triggered phagocyte cell death involves the uptake of dAdo and dGuo via human equilibrative nucleoside transporter 1 (hENT1), intracellular phosphorylation of dAdo and dGuo by adenosine kinase (ADK) and deoxycytidine kinase (dCK), and subsequent imbalanced expansion of intracellular dATP and dGTP pools which trigger caspase-3-dependent immune cell death (*Winstel et al., 2018*; *Tantawy et al., 2022*). Of note, genetic disruption of this metabolic cascade rendered macrophages resistant to AdsA-derived death-effector deoxyribonucleosides, leading to the accumulation of phagocytes within the deeper cavity of abscesses and accelerated clearance of staphylococci *in vivo* (*Winstel et al., 2018*; *Winstel et al., 2019*; *Tantawy et al., 2022*). Thus, elements of the mammalian purine salvage pathway represent attractive targets for the design of novel host-directed therapeutics that aim at boosting macrophage survival during acute and persistent *S. aureus* infections. In particular, pharmacological inhibition of dCK by small molecule inhibitors may terminate the intracellular conversion of *S. aureus*- and AdsA-derived death-effector deoxyribonucleosides into deoxyribonucleoside monophosphates

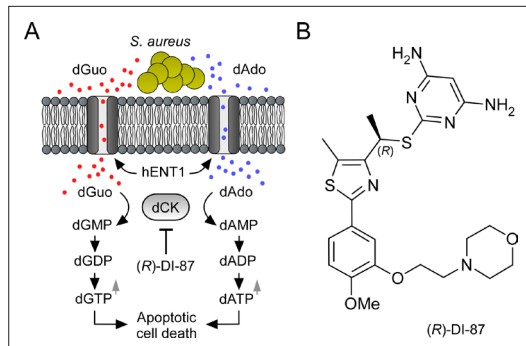

**Figure 1.** Mode of action of the dCK-specific inhibitor (*R*)-DI-87. (**A**) Scheme illustrating the mode of action of the dCK-specific inhibitor (*R*)-DI-87. *S. aureus*-derived dAdo and dGuo are pumped into phagocytes via human equilibrative transporter 1 (hENT1). Deoxycytidine kinase (dCK) converts dAdo and dGuo into appropriate deoxyribonucleoside monophosphates thereby triggering an accumulation of apoptosis-stimulating deoxyribonucleoside di- and triphosphates. (*R*)-DI-87 interferes with this pathway by inhibiting dCK, thus preventing host cell death. (**B**) Structure of (*R*)-DI-87.

and subsequent accumulation of apoptosis-triggering deoxyribonucleoside triphosphates in host phagocytes (*Figure 1A*). If so, *S. aureus*- and dAdo/dGuo-mediated macrophage cell death may efficiently be prevented thereby aiding in the clearance of staphylococci during abscess formation and persistent infections.

Here, we provide proof of this concept by establishing a novel host-directed therapeutic strategy for prophylaxis and therapy of staphylococcal infectious diseases. Specifically, we show that the dCK-specific small molecule inhibitor (*R*)-DI-87 (((*R*)–2-((1-(2-(4-methoxy-3-(2-morpholinoethoxy)phenyl)–5-methylthiazol-4-yl)ethyl)thio)pyrimidine-4,6-diamine)), an orally active, well-tolerated clinical-stage anticancer drug (*Figure 1B*; *Poddar et al., 2020*), protects host phagocytes from staphylococcal death-effector deoxyribonucleoside-mediated cytotoxicity and caspase-3-dependent cell death. Our results further suggest that administration of (*R*)-DI-87 boosts macrophage survival during abscess formation and thus improves clinical outcomes in *S. aureus*-infected laboratory animals.

## Results

### (*R*)-DI-87-mediated blockade of mammalian dCK protects host immune cells from staphylococcal death-effector deoxyribonucleosides

To analyze whether pharmacological inhibition of mammalian dCK may represent a suitable strategy to protect host phagocytes from staphylococcal death-effector deoxyribonucleosides, we initially took advantage of various tissue culture model systems and pre-incubated human U937 monocytes (U937) or U937-derived macrophages (U937 MΦ) with (*R*)-DI-87. Controls received vehicle only. Following pre-incubation, U937 or U937 MΦ were exposed to dAdo or dGuo and analyzed for viability rates 48 hr post-intoxication. Of note, (*R*)-DI-87-mediated inhibition of mammalian dCK in U937 or U937 MΦ efficiently prevented dAdo- or dGuo-induced cell death in a dose-dependent manner (*Figure 2A–D* and *Figure 2—figure supplement 1A–D*). Moreover, these effects resembled the phenotype of dCK-deficient U937 or U937 MΦ, which were found to be refractory to dAdo- or dGuo-mediated cytotoxicity (*Figure 2A–D*; *Winstel et al., 2018*; *Tantawy et al., 2022*). In addition, we observed that pre-treatment of primary human CD14⁺ monocytes or human monocyte-derived macrophages (HMDMs) with (*R*)-DI-87 blocked dAdo- or dGuo-triggered cytotoxicity suggesting that dCK inhibition may suppress the toxigenic properties of staphylococcal death-effector deoxyribonucleosides (*Figure 2E–H*). To test this conjecture, we expressed a soluble and affinity-tagged recombinant form of *S. aureus* AdsA (hereafter termed rAdsA) in *Escherichia coli*, which was purified, released of its affinity tag, and subsequently used for cytotoxicity assays. Particularly, we incubated rAdsA with either dAMP or dGMP according to a published protocol (*Tantawy et al., 2022*) and added the resulting, dAdo- or dGuo-containing and filter-sterilized reaction products to U937 or U937 MΦ that received (*R*)-DI-87 or were left untreated prior to intoxication. While enzymatic reactions that contained rAdsA and purine deoxyribonucleoside monophosphates triggered cell death of human monocytes or macrophages in this approach, (*R*)-DI-87-exposed cells were fully protected against the cytotoxic effect of dAdo and dGuo further indicating that inhibition of mammalian dCK helps to neutralize AdsA-derived death-effector deoxyribonucleosides (*Figure 2I–J*). In this regard, we also wondered whether these results can be recapitulated by using live bacteria and took advantage of previously described methodologies (*Thammavongsa et al., 2013*; *Winstel et al., 2018*; *Winstel et al., 2019*; *Tantawy et al., 2022*). More precisely, we incubated wild-type *S. aureus* Newman or its

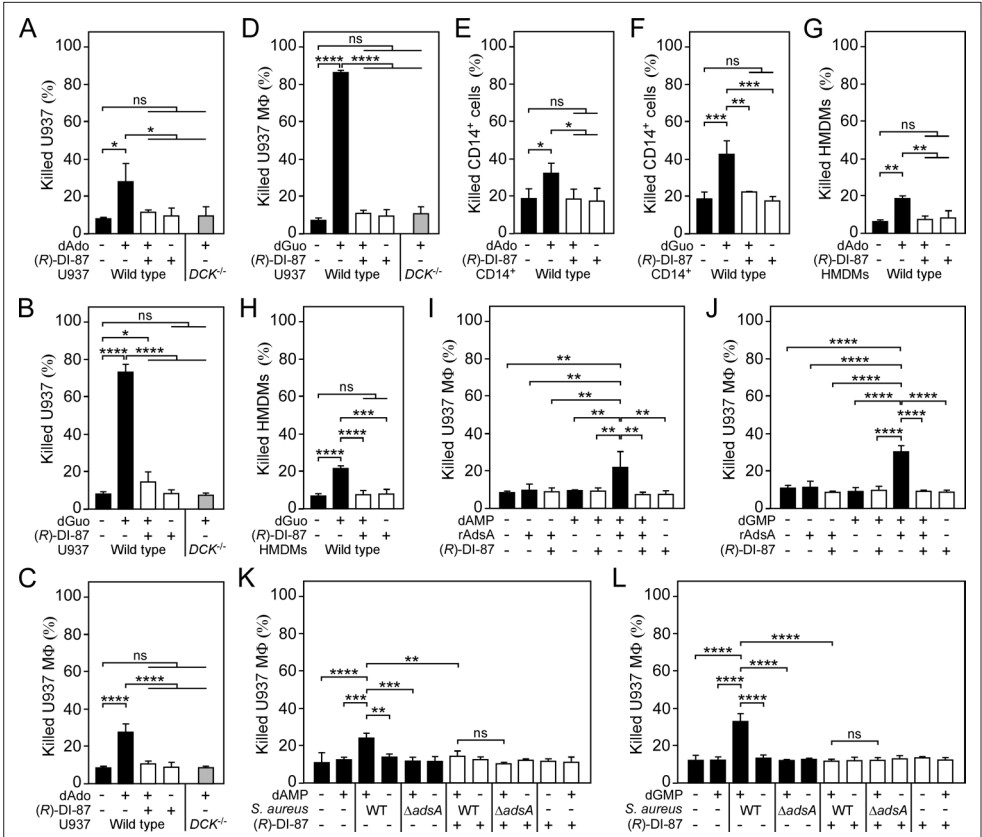

**Figure 2.** (*R*)-DI-87 protects phagocytes from death-effector deoxyribonucleoside-mediated cytotoxicity. (**A–D**) Survival rates of human U937 monocyte-like cells (U937) (**A, B**) or U937-derived macrophages (U937 MΦ) (**C, D**) exposed to dAdo or dGuo in the presence (+) or absence (-) of 1 µM (*R*)-DI-87. Cells were also exposed to the inhibitor or vehicle only. U937 *DCK−/−* were included as a control. (**E–H**) Survival rates of human CD14+ monocytes (**E, F**) or human monocyte-derived macrophages (HMDMs) (**G, H**) exposed to dAdo or dGuo in the presence (+) or absence (-) of 1 µM (*R*)-DI-87. Cells were also exposed to the inhibitor or vehicle only. (**I–J**) Survival rates of U937 MΦ exposed to rAdsA-derived dAdo (**I**) or dGuo (**J**). rAdsA was incubated with dAMP or dGMP and reaction products containing dAdo or dGuo were used to treat phagocytes in the presence (+) or absence (-) of 1 µM (*R*)-DI-87. Controls lacked rAdsA or deoxyribonucleoside monophosphates, or included reaction buffer only as indicated with + and − symbols. (**K, L**) Survival of vehicle- (-) or (*R*)-DI-87-exposed (+) U937 MΦ after treatment with culture medium (RPMI) that had been conditioned by incubation with either wild-type *S. aureus* Newman (WT) or its *adsA* mutant (Δ*adsA*) in the presence or absence of dAMP (**K**) or dGMP (**L**) as indicated with + and – symbols. Controls are indicated. 100 µM (**A–B**; **E–F**) or 200 µM (**C-D**; **G–H**) of dAdo or dGuo were used to treat the cells. Cell survival rates were analyzed 48 hr (**A–J**) or 24 h (**K, L**) post-treatment. Data are the mean (± standard deviation [SD]) values from at least three independent determinations. Primary cell experiments include at least three independent donors. Statistically significant differences were analyzed by two-way (**A–D**) or one-way (**E–L**) analysis of variance (ANOVA) followed by Tukey's multiple-comparison test; ns, not significant (*P*≥0.05); *, p<0.05; **, p<0.01; ***, p<0.001; ****, p<0.0001.

The online version of this article includes the following source data and figure supplement(s) for figure 2:

**Source data 1.** Data used to generate *Figure 2*.

**Figure supplement 1.** (*R*)-DI-87 prevents death-effector deoxyribonucleoside-triggered immune cell death in a dose-dependent manner.

**Figure supplement 1—source data 1.** Data used to generate *Figure 2—figure supplement 1*.

---

*adsA* variant in the presence or absence of purine deoxyribonucleoside monophosphates (dAMP or dGMP) to obtain conditioned culture media, which were filter-sterilized and added to vehicle- or (*R*)-DI-87-exposed human U937 macrophages. In line with earlier studies (*Thammavongsa et al., 2013*; *Winstel et al., 2018*; *Winstel et al., 2019*; *Tantawy et al., 2022*), macrophage killing required purine deoxyribonucleoside monophosphate-conditioned media and *adsA*-proficient staphylococci as only

wild-type *S. aureus* Newman and dAMP- or dGMP-supplemented media triggered phagocyte cell death in this approach (*Figure 2K–L*). However, (*R*)-DI-87-exposed cells could not be killed in these experiments supporting the idea that pharmacological inhibition of host dCK can shield macrophages from *S. aureus*- and AdsA-driven cell death (*Figure 2K–L*). Collectively, these initial data demonstrate that (*R*)-DI-87 is a suitable small molecule dCK inhibitor capable of preventing host immune cell death induced by toxigenic products of the staphylococcal Nuc/AdsA pathway.

## (*R*)-DI-87 prevents death-effector deoxyribonucleoside-induced activation of apoptosis

Earlier work demonstrated that staphylococcal dAdo and dGuo target the mammalian purine salvage pathway to trigger an exaggerated biogenesis of deoxyribonucleoside triphosphates, thereby igniting caspase-3-dependent host immune cell death (*Winstel et al., 2018*; *Winstel et al., 2019*; *Tantawy et al., 2022*). Since the accumulation of deoxyribonucleoside triphosphates and associated apoptotic signaling via cleavage of caspase-3 exclusively occurs in dCK-proficient monocytes

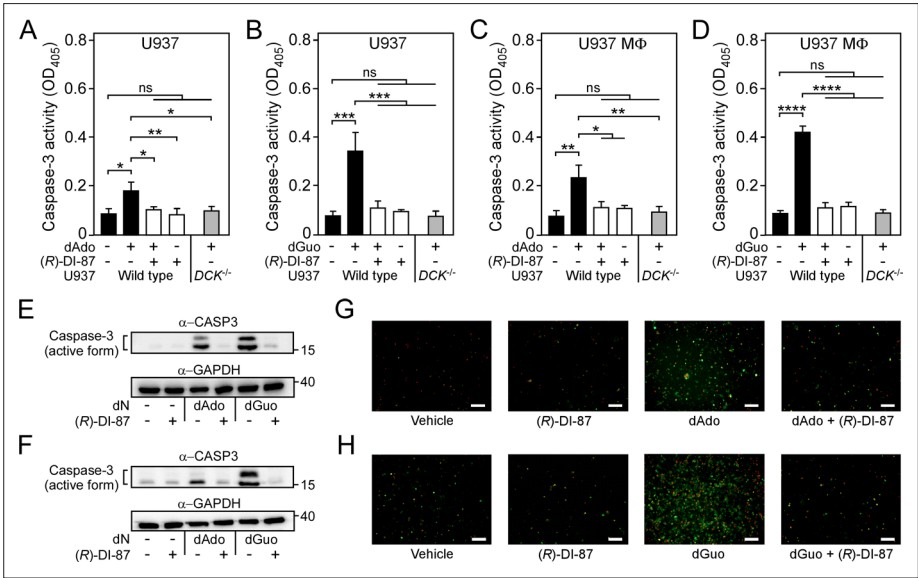

**Figure 3.** Selective inhibition of dCK prevents death-effector deoxyribonucleoside-mediated induction of immune cell apoptosis. (**A–D**) Analysis of caspase-3 activity in human U937 monocyte-like cells (U937) (**A, B**) or U937-derived macrophages (U937 MΦ) (**C, D**) exposed to dAdo or dGuo in the presence (+) or absence (-) of 1 µM (*R*)-DI-87. Cells were also exposed to the inhibitor or vehicle only. U937 *DCK⁻/⁻* were included as a control. Caspase-3 activity was analyzed using a colorimetric assay. (**E–F**) Immunoblotting of lysates obtained from U937 (**E**) or U937 MΦ (**F**) exposed to dAdo or dGuo in the presence (+) or absence (-) of 1 µM (*R*)-DI-87. Controls are indicated (+/– symbols). A specific antibody was used that can also detect the cleaved (active) form of caspase-3 (α-CASP3). GAPDH was used as a loading control (α-GAPDH). Numbers to the right of blots indicate the migration of molecular weight markers in kilodaltons. (**G–H**) Analysis of (*R*)-DI-87-dependent prevention of host cell apoptosis via immunofluorescence microscopy. U937 MΦ were exposed to dAdo (**G**) or dGuo (**H**) in the presence or absence of 1 µM (*R*)-DI-87 and stained using FITC-annexin-V/PI. Controls are indicated. Scale bars depict a length of 100 µm. Representative blots and images are shown. 100 µM (**A–B**; **E**) or 200 µM (**C-D**; **F–H**) of dAdo or dGuo were used to treat the cells. Apoptosis rates were analyzed 24 hr post-treatment. Data are the mean (± standard deviation [SD]) values from three independent determinations. Statistically significant differences were analyzed by two-way analysis of variance (ANOVA) followed by Tukey's multiple-comparison test; ns, not significant (p≥0.05); *, p<0.05; **, p<0.01; ***, p<0.001; ****, p<0.0001.

The online version of this article includes the following source data and figure supplement(s) for figure 3:

**Source data 1.** Original and unedited western blot scans used to generate *Figure 3E–F*.

**Source data 2.** PDF containing uncropped and labeled western blot scans used to generate *Figure 3E–F*.

**Source data 3.** Data used to generate *Figure 3*.

**Figure supplement 1.** Inhibition of dCK prevents death-effector deoxyribonucleoside-mediated induction of immune cell apoptosis in primary human macrophages.

or macrophages upon intoxication with death-effector deoxyribonucleosides such as dAdo (*Winstel et al., 2018*; *Tantawy et al., 2022*), it seemed plausible to us at this stage that administration of (*R*)-DI-87 and inhibition of dCK may suppress dAdo- or dGuo-mediated activation of the programmed cell death machinery. To test this hypothesis, U937 or U937 MΦ were pre-incubated with (*R*)-DI-87, exposed to dAdo or dGuo, and used to generate cell extracts for evaluation of caspase-3 activity by measuring the hydrolysis of the caspase-3-specific peptide substrate Ac-DEVD-pNA. As expected, (*R*)-DI-87-treatment of U937 or U937 MΦ significantly decreased dAdo- or dGuo-mediated activation of caspase-3 activity (*Figure 3A–D*). Since caspase-3 represents the key modulator of the apoptosis signaling pathway, these data suggest that (*R*)-DI-87-mediated dCK inhibition selectively prevented dAdo- or dGuo-mediated induction of apoptotic cell death in host immune cells (*Figure 3A–D*). To verify these results further, cell extracts were also probed with a specific antibody capable of detecting the inactive pro-form and cleaved (active) form of human caspase-3. In agreement with the enzymatic activity assay, (*R*)-DI-87-treatment of U937 or U937 MΦ prevented caspase-3 activation (*Figure 3E–F*). Moreover, we exposed U937 phagocytes to (*R*)-DI-87 and assessed dAdo- or dGuo-mediated activation of apoptotic signaling via immunofluorescence microscopy. Microscopy-based analysis of (*R*)-DI-87- and death-effector deoxyribonucleoside-exposed macrophages confirmed that (*R*)-DI-87 inhibited activation of programmed cell death and apoptosis as positive signals for annexin-V/PI were strongly decreased in samples that have been exposed to the dCK inhibitor (*Figure 3G–H*). Of note, similar findings were also obtained with primary HMDMs, suggesting that (*R*)-DI-87 is a suitable agent to block death-effector deoxyribonucleoside-mediated induction of apoptosis in host phagocytes (*Figure 3—figure supplement 1*). Together, these data indicate that administration of (*R*)-DI-87 and associated inhibition of mammalian dCK prevents apoptotic cell death in phagocytes caused by *S. aureus* AdsA-derived death-effector deoxyribonucleosides.

## Pharmacological inhibition of host dCK diminishes *S. aureus* abscess formation in a mouse model of bloodstream infection

To evaluate the therapeutic value of (*R*)-DI-87 in live animals, initial *in vivo* experiments aimed at analyzing the safety of (*R*)-DI-87 in mice following continuous dCK inhibitor treatment. Thus, cohorts of female C57BL/6 mice were treated from day 0 onwards with either vehicle (40% Captisol) or (*R*)-DI-87 (75 mg/kg) via oral gavage in 12 hr intervals according to a published protocol (*Chen et al., 2023*). On day 16, peripheral blood was collected from both cohorts of mice and subjected to a FACS-based immuno-phenotyping approach (*Figure 4A*). Continuous treatment of animals with (*R*)-DI-87 did not alter the immune cell composition of peripheral blood (*Figure 4A*). Likewise, endpoint analysis (day 23) uncovered no developmental errors or differences in lymphocyte development as immune cell profiles of spleen tissues along with organ cellularity or mouse body weights were unaffected by (*R*)-DI-87 treatment (*Figure 4B–D* and *Figure 4—figure supplement 1*). Further, administration of (*R*)-DI-87 to mice did not cause any other obvious phenotype during this procedure but led to the accumulation of deoxycytidine, the natural substrate of dCK (*Reichard, 1988*; *Arnér and Eriksson, 1995*), in plasma suggesting that (*R*)-DI-87 not only represents a safe but also highly selective inhibitor of host dCK (*Figure 4E*). Next, we sought to investigate whether (*R*)-DI-87-mediated protection of mammalian phagocytes may represent a valuable strategy to prevent staphylococcal diseases and tested the therapeutic efficacy of the compound in a mouse model of *S. aureus* bloodstream infection. In this model, C57BL/6 mice were treated with (*R*)-DI-87 (75 mg/kg) or vehicle (40% Captisol) via oral gavage every 12 hr and challenged with a single dose of 1.0 x 10^7 CFU *S. aureus* Newman, a human clinical isolate (*Duthie and Lorenz, 1952*). Five days post-infection, mice were euthanized. Livers and kidneys were dissected and analyzed for visible abscess lesions. Subsequently, tissue homogenates were prepared and plated on agar plates to measure bacterial loads in infected organs. Notably, abscess numbers and bacterial loads were significantly reduced in livers and kidneys in (*R*)-DI-87-treated animals as compared to control mice validating (*R*)-DI-87 as a novel host-directed and anti-infective drug that protects against staphylococcal abscess formation (*Figure 4F–I*). To test whether *S. aureus* exploits the activity of dCK along with the purine salvage pathway during infection, cohorts of mice were also challenged with the *S. aureus* Newman *adsA* mutant. (*R*)-DI-87-treated animals no longer displayed increased resistance to *S. aureus* infection (*Figure 4F–I*). Moreover, infection with *S. aureus adsA*-deficient bacteria phenocopied (*R*)-DI-87-mediated inhibition of dCK, in line with the concept that AdsA is required for establishing persistent infections in host tissues

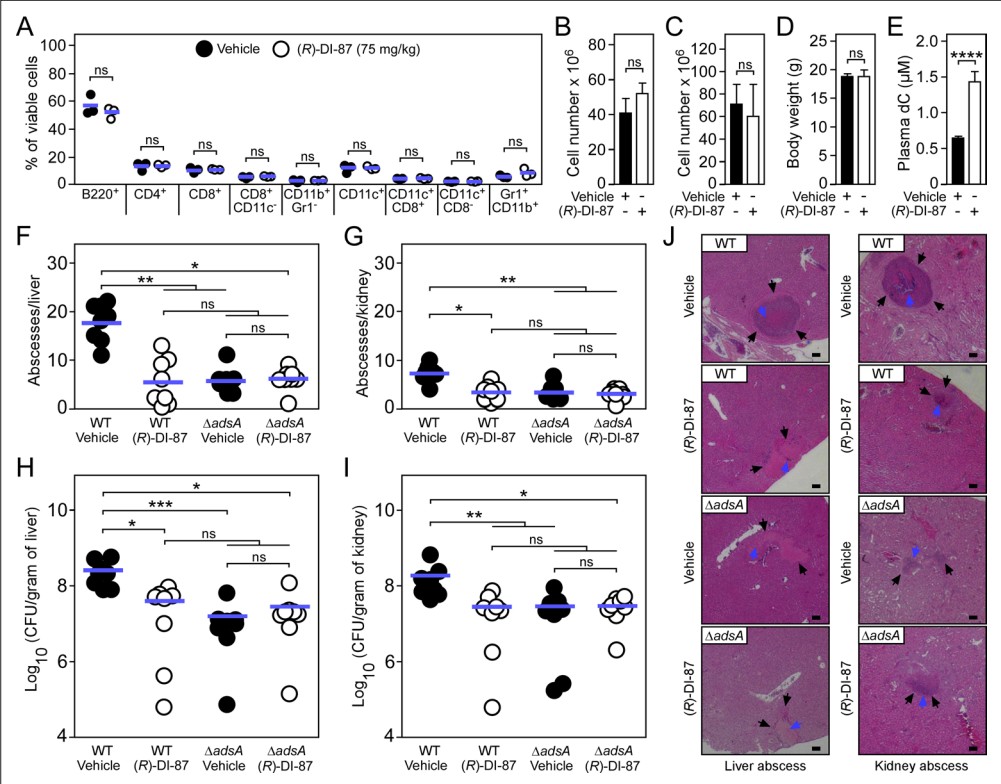

**Figure 4.** (R)-DI-87 protects against *S. aureus* invasive disease. (**A–D**) Safety assessment of (R)-DI-87 in mice. Cohorts of female C57BL/6 mice were treated with (R)-DI-87 (75 mg/kg) or vehicle (40% Captisol) via oral gavage in 12 hr intervals for 23 days. On day 16, peripheral blood was collected and subjected to a FACS-based immuno-phenotyping approach (**A**). Subsequent panels indicate the cellularity of spleen (**B**) and thymus (**C**) tissues along with the body weight of mice (**D**) on day 23. (**E**) Analysis of deoxycytidine (dC) content in mouse plasma following continuous dCK inhibitor treatment on day 23. (**F–I**) Enumeration of visible surface abscesses and staphylococcal loads in organs of *S. aureus*-challenged C57BL/6 mice treated with (R)-DI-87 (75 mg/kg) or vehicle (40% Captisol). Mice received (R)-DI-87 (75 mg/kg) or vehicle (40% Captisol) via oral gavage every 12 hr and were challenged with $10^7$ CFU of wild-type *S. aureus* Newman (WT) or its *adsA* mutant ($\Delta adsA$). Data for female C57BL/6 mice are displayed (n=8). Bacterial burden was enumerated as $\log_{10}$ CFU per gram of tissue at 5 days post-infection. Horizontal blue bars represent the mean values of visible abscesses per organ (**F–G**) or indicate the mean CFU count in each cohort (**H–I**). (**J**) Microscopic images of H&E–stained liver or renal tissues obtained after necropsy of *S. aureus*-challenged C57BL/6 mice treated with (R)-DI-87 (75 mg/kg) or vehicle (40% Captisol). Mice received (R)-DI-87 (75 mg/kg) or vehicle (40% Captisol) via oral gavage every 12 hr and were challenged with $10^7$ CFU of wild-type *S. aureus* Newman (WT) or its *adsA* mutant ($\Delta adsA$). Arrows point to immune cell infiltrates (black) or replicating staphylococci (blue). Scale bars depict a length of 100 µm. Representative images are shown. Statistically significant differences were analyzed by a two-tailed Student's t-test (**A–E**) or with the Kruskal–Wallis test corrected with Dunn's multiple comparison (**F–I**). ns, not significant (p≥0.05); *, p<0.05; **, p < 0.01; ***, p<0.001; ****, p<0.0001.

The online version of this article includes the following source data and figure supplement(s) for figure 4:

**Source data 1.** Data used to generate *Figure 4*.

**Figure supplement 1.** Immuno-phenotypic assessment of murine spleen tissues following continuous dCK inhibitor treatment.

**Figure supplement 1—source data 1.** Data used to generate *Figure 4—figure supplement 1*.

**Figure supplement 2.** (R)-DI-87 does not interfere with staphylococcal survival in blood.

**Figure supplement 2—source data 1.** Data used to generate *Figure 4—figure supplement 2*.

(**Figure 4F–I**). In light of these findings, infected organs were also fixed with formalin, embedded into paraffin, thin-sectioned, and analyzed for histopathology. As expected, histopathological analysis of hematoxylin and eosin (H&E)-stained liver or renal tissues revealed that wild-type *S. aureus* Newman formed structured abscesses in vehicle-treated mice (**Figure 4J**). However, lesions obtained from (*R*)-DI-87-treated animals appeared smaller in size and typically did not harbor a discernable organization of staphylococci, further demonstrating that administration of (*R*)-DI-87 represents a medically valuable strategy to mitigate *S. aureus* abscess formation (**Figure 4J**). Presumably, (*R*)-DI-87-treatment protects macrophages against AdsA-derived death-effector deoxyribonucleosides and therefore enhances phagocyte survival during abscess formation as (*R*)-DI-87 neither improved the killing of *S. aureus* in human or mouse blood, nor it displayed antimicrobial activity (**Figure 4—figure supplement 2A–B** and **Supplementary file 1**). Collectively, these data indicate that inhibition of mammalian dCK by (*R*)-DI-87 attenuates *S. aureus* abscess formation and disease pathogenesis *in vivo*.

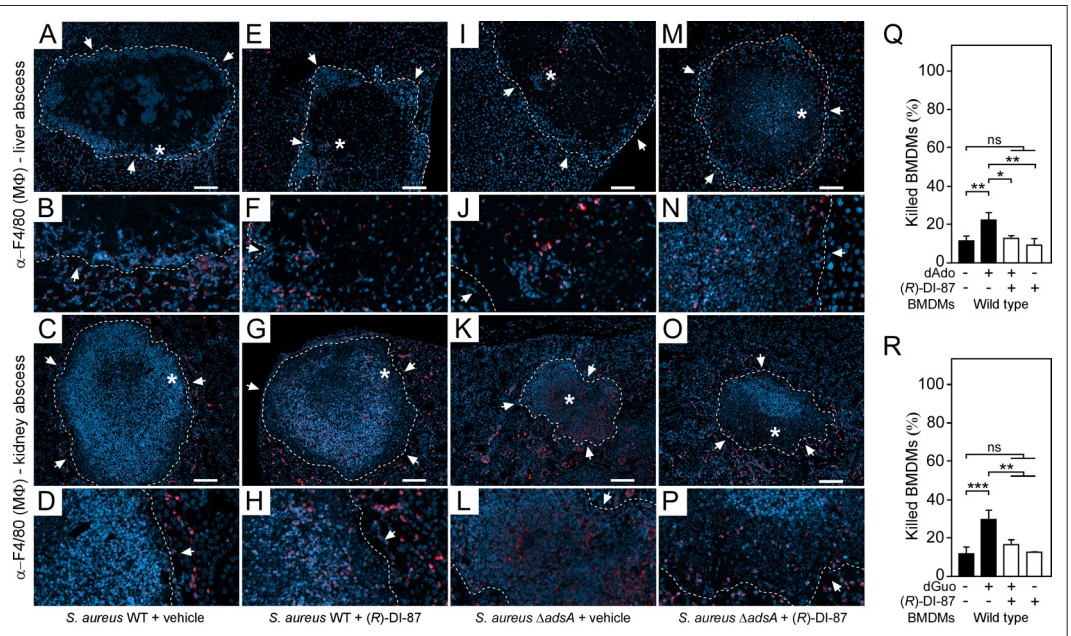

**Figure 5.** (*R*)-DI-87-mediated inhibition of dCK enhances macrophage infiltration into staphylococcal abscesses. (**A–P**) Immunofluorescence microscopy-based detection of macrophages in liver or renal tissues isolated 5 days after intravenous injection of 10⁷ CFU of wild-type *S. aureus* Newman (WT) (**A–H**) or its *adsA* mutant (Δ*adsA*) (**I–P**) into female C57BL/6 mice treated with (*R*)-DI-87 (75 mg/kg) or vehicle (40% Captisol). Mice received (*R*)-DI-87 (75 mg/kg) or vehicle (40% Captisol) via oral gavage every 12 hr. White arrows point at the periphery of infectious foci (dashed lines). Magnifications of lesions from upper panels are indicated. Asterisk symbols define the region enlarged in the magnification counterpart images. Thin sections were stained with α-F4/80 antibodies (macrophages; red). Nuclei were labeled with DAPI (blue). Scale bars shown in the upper panels depict 100 μm length. Representative images are shown. (**Q, R**) Survival rates of female mice-derived bone marrow-derived macrophages (BMDMs) exposed to dAdo (**Q**) or dGuo (**R**) in the presence (+) or absence (-) of 1 μM (*R*)-DI-87. Cells were also exposed to the inhibitor or vehicle only. 200 μM of dAdo or dGuo were used to treat the cells. Cell survival rates were analyzed 48 hr post-treatment. Data are the mean (± standard deviation [SD]) values from three independent determinations. Statistically significant differences were analyzed by one-way analysis of variance (ANOVA) followed by Tukey's multiple-comparison test; ns, not significant (p≥0.05); *, p<0.05; **, p<0.01; ***, p<0.001.

The online version of this article includes the following source data and figure supplement(s) for figure 5:

**Source data 1.** Data used to generate **Figure 5**.

**Figure supplement 1.** (*R*)-DI-87-mediated inhibition of dCK shields male mice-derived phagocytes from death-effector deoxyribonucleosides.

**Figure supplement 1—source data 1.** Data used to generate **Figure 5—figure supplement 1**.

## (*R*)-DI-87-treatment amplifies macrophage infiltration into staphylococcal infectious foci

Previous studies revealed that macrophages with defects in the purine salvage pathway-apoptosis axis are refractory to staphylococcal dAdo or dGuo (*Winstel et al., 2018*; *Winstel et al., 2019*; *Tantawy et al., 2022*). As a result, death-effector deoxyribonucleoside-resistant tissue macrophages were found to accumulate within deep-seated abscesses of *S. aureus*-challenged mice, a phenomenon that contributed to accelerated clearance of staphylococci (*Winstel et al., 2019*). Thus, we hypothesized that pharmacological inhibition of host dCK might augment phagocyte infiltration into *S. aureus*-derived abscesses, thereby explaining reduced bacterial burdens in (*R*)-DI-87-treated laboratory animals. To pursue this possibility, we established an immunofluorescence microscopy-based approach to detect macrophages in liver or renal tissues of *S. aureus* Newman wild type-infected C57BL/6 mice that were treated with (*R*)-DI-87 or vehicle only. As expected, immunofluorescence staining of tissue abscesses obtained from vehicle-treated animals revealed that F4/80-positive macrophages resided at the periphery of infectious foci (*Figure 5A–D*). On the contrary, lesions derived from mice that received (*R*)-DI-87 differed as they contained infiltrates of F4/80-positive macrophages within the neutrophil cuff, suggesting that (*R*)-DI-87-mediated inhibition of host dCK terminates macrophage exclusion from staphylococcal infectious foci (*Figure 5E–H*). To further delineate whether these results correlate with AdsA and staphylococcal death-effector deoxyribonucleoside-mediated manipulation of host dCK in phagocytes, tissues of *S. aureus* Newman *adsA* mutant-infected mice, which received either (*R*)-DI-87 or vehicle, were also examined via immunofluorescence microscopy and analyzed for the presence of F4/80-positive macrophages. Of note, *adsA* mutant-derived tissue lesions were also characterized by increased infiltration rates of F4/80-positive phagocytes, irrespectively of whether animals were treated with (*R*)-DI-87 or vehicle during the course of the infection (*Figure 5I–P*). Thus, (*R*)-DI-87-mediated blockade of host dCK phenocopied the *adsA* mutation in *S. aureus*. Lastly, (*R*)-DI-87- or vehicle-treated bone-marrow-derived macrophages (BMDMs) that were isolated from female animals were analyzed for survival rates upon exposure to dAdo or dGuo. Compared to vehicle-treated cells, which were susceptible to death-effector deoxyribonucleosides, BMDMs exposed to (*R*)-DI-87 displayed increased resistance toward dAdo and dGuo presumably explaining their abundance in infectious foci obtained from wild-type *S. aureus* Newman-challenged and dCK inhibitor-treated mice (*Figure 5Q–R*). Similar findings were also obtained for male animal-derived BMDMs suggesting that sex might not impact the efficacy of (*R*)-DI-87 in laboratory animals (*Figure 5—figure supplement 1A–B*). In summary, these data suggest that (*R*)-DI-87-treatment protects macrophages from staphylococcal death-effector deoxyribonucleosides and therefore boosts their infiltration into persistent abscesses in organ tissues (*Figure 6*).

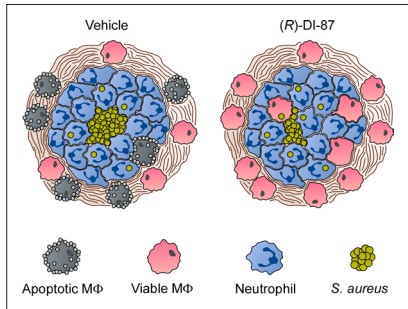

**Figure 6.** Proposed model of (*R*)-DI-87-mediated protection of phagocytes during *S. aureus* abscess formation. Diagram illustrating the (*R*)-DI-87-mediated protection of macrophages during the development of staphylococcal abscesses. While phagocytes get killed by *S. aureus*-derived death-effector deoxyribonucleosides in vehicle-treated animals, (*R*)-DI-87 protects macrophages and boosts their infiltration into the deeper cavity of infectious foci thereby enhancing eradication of staphylococci.

## Discussion

Bacterial infectious diseases are typically treated with specific or broad-spectrum antibiotics. However, many microbial pathogens have developed resistance mechanisms against these drugs and other antimicrobial agents either via mutational changes within the drug target or by importing resistance genes from other bacteria (*Klevens et al., 2008*; *Yaw et al., 2014*; *Nübel, 2016*; *Jian et al., 2021*; *GBD 2019 Antimicrobial Resistance Collaborators, 2022a*; *GBD 2019 Antimicrobial Resistance Collaborators, 2022a*). Accordingly, infections by multidrug-resistant microbes are frequently associated with failure of therapy and increased mortality in both, industrialized and developing countries around the world (*Klevens et al., 2008*; *Yaw et al., 2014*; *GBD 2019 Antimicrobial Resistance Collaborators, 2022a*; *GBD 2019 Antimicrobial*

*Resistance Collaborators, 2022b*). Each year, several million people die due to infections caused by drug-resistant bacteria such as *Mycobacterium tuberculosis*, *E. coli*, or *S. aureus* underscoring the very urgent need of new therapeutics and anti-infective compounds against which bacteria cannot develop resistance (*Pai et al., 2016*; *GBD 2019 Antimicrobial Resistance Collaborators, 2022a*; *GBD 2019 Antimicrobial Resistance Collaborators, 2022b*).

Host-directed therapy is an emerging therapeutic concept that eliminates the possibility of drug resistance development in microbial pathogens (*Kaufmann et al., 2018*). This innovative approach aims at targeting specific host determinants and signaling cascades bacterial or other pathogenic microorganisms may attack or exploit to establish acute and persistent infections (*Kaufmann et al., 2018*). Overall, this strategy seeks to ameliorate host immunity, immune cell survival, and immunopathology, as well as pathogen control without interfering with microbial replication, growth, or metabolism (*Kaufmann et al., 2018*). By applying this concept along with a drug-repurposing approach, we here report that administration of (*R*)-DI-87, a soluble, orally active, and high-affinity inhibitor of host dCK (*Poddar et al., 2020*), reduces *S. aureus* abscess formation in a mouse model of bloodstream infection. In particular, we show that (*R*)-DI-87-mediated blockade of host dCK prevents staphylococcal death-effector deoxyribonucleoside-triggered apoptotic cell death in phagocytes of human and animal origin. Accordingly, oral administration of (*R*)-DI-87 in mice rendered host phagocytes refractory to AdsA-derived dAdo and dGuo thereby leading to an accumulation of *S. aureus*-eliminating macrophages within deeper cavities of infectious foci along with accelerated phagocytic clearance of staphylococci. Since dCK is a rate-limiting enzyme of the nucleoside salvage pathway that catalyzes the intracellular conversion of deoxyribonucleosides into deoxyribonucleoside monophosphates (*Arnér and Eriksson, 1995*), (*R*)-DI-87-mediated inhibition of dCK most likely suppresses an uncontrolled biogenesis of deoxyribonucleoside di- and triphosphates that may trigger replication errors and apoptosis in host cells. Consistent with this model, earlier work suggested that dCK deficiency in phagocytes or other cells such as lymphoblasts prevents an overload of intracellular deoxyribonucleoside triphosphates upon exposure to death-effector deoxyribonucleosides (*Gudas et al., 1978*; *Winstel et al., 2018*). Thus, inhibition of host dCK represents an attractive host-directed therapeutic intervention strategy that terminates a refined immuno-evasive maneuver *S. aureus* has evolved to establish persistent infections in mammalian hosts. The technological advantage of this therapeutic approach, beyond the clinical importance and long-term economic benefits for the medical-therapeutic sector, is undoubtedly the exploitation of an already existing, highly selective, and clinical-stage compound that can easily be administered to dampen metastatic abscess formation in organs upon staphylococcal bloodstream infection. In particular, (*R*)-DI-87 does not exhibit detrimental side effects as (*R*)-DI-87-exposed laboratory animals neither lost weight during long-term treatment nor displayed any other adverse effects that may negatively affect host immune responses. Specifically, the immune cell composition profile of mice that received (*R*)-DI-87 was not altered when compared to mice that received the vehicle control. This suggests that patrolling neutrophils known to play a significant role during staphylococcal infections (*Spaan et al., 2013*) along with other crucial elements of the host immune system are fully functional in (*R*)-DI-87-treated hosts, thereby offering tremendous opportunities to improve infection outcomes in hospitalized or critically ill patients. Further, usage of (*R*)-DI-87 represents a host-directed and anti-infective therapeutic approach to which *S. aureus* and other multidrug-resistant bacteria cannot develop resistance mechanisms. In this regard, we note that many other medically highly relevant bacterial pathogens including streptococci, *Bacillus anthracis*, and multidrug-resistant *Staphylococcus pseudintermedius* synthesize effector-nucleosides in order to modulate host immune cell responses (*Thammavongsa et al., 2009*; *Liu et al., 2014*; *Zheng et al., 2015*; *Ma et al., 2017*; *Dai et al., 2018*; *Bünsow et al., 2021*). Thus, (*R*)-DI-87-mediated inhibition of dCK may also help to protect host cells from apoptotic cell death or other adverse effects during other bacterial diseases in human or animal hosts. Even polymicrobial infections caused by various microbial pathogens and other infectious agents that manipulate the purine salvage pathway-apoptosis axis during infection can eventually be prevented, attenuated, or treated by using (*R*)-DI-87. Nevertheless, administration of (*R*)-DI-87 did not completely abolish disease development and abscess formation during sub-lethal bloodstream infection in mice. Thus, double-pronged therapies which cover the usage of (*R*)-DI-87 combined with antibiotics, various anti-infective agents, or monoclonal antibodies, some of which are known to neutralize predominant staphylococcal virulence determinants, including secreted toxins (*Ragle and Bubeck Wardenburg, 2009*; *Hua et al., 2014*; *Thomsen et al., 2017*),

protein A (*Kim et al., 2012*; *Chen et al., 2022*), and specific surface antigens (*Bennett et al., 2019*; *de Vor et al., 2022*) along with usage of other host-directed therapeutics that impede *S. aureus* disease pathogenesis (*Bravo-Santano et al., 2019*; *Alphonse et al., 2021*), may essentially improve disease progression during staphylococcal infections. Ultimately, combinatorial therapies that involve usage of (*R*)-DI-87 may even help to enhance outcomes of life-threatening septicemia or other fatal bacterial infectious diseases and should therefore be considered for pre-clinical tests in animal models.

Overall, (*R*)-DI-87-mediated inhibition of dCK is a novel host-directed therapeutic concept to mitigate staphylococcal abscess formation in organ tissues upon bloodstream infection. Since (*R*)-DI-87 is extremely well tolerated in humans (Kenneth A. Schultz, Trethera Corporation, Los Angeles, CA, USA; personal communication, 2023) and currently being investigated in phase I clinical trials for the treatment of non-communicable diseases, including cancer and multiple sclerosis (*Poddar et al., 2020*; *Chen et al., 2023*), this compound already meets the requirements for a prompt launch of clinical studies in individuals that suffer from local or invasive staphylococcal infections. Optimization of (*R*)-DI-87 or any medically acceptable derivatives thereof may further accelerate this process and concurrently could also aid in the design of pre-exposure prophylactic agents for hospitalized or high-risk patients, with the primary aim to enhance infection control and public health.

## Materials and methods

**Key resources table**

| Reagent type (species) or resource | Designation | Source or reference | Identifiers | Additional information |
|---|---|---|---|---|
| Strain, strain background (*Staphylococcus aureus* Newman) | *S. aureus* Newman wild type | *Duthie and Lorenz, 1952* | N/A | |
| Strain, strain background (*S. aureus* Newman Δ*adsA*) | *S. aureus* Newman Δ*adsA* | *Tantawy et al., 2022* | N/A | |
| Strain, strain background (*Escherichia coli* BL21 (DE3) pGEX-2T-*adsA*) | *E. coli* BL21 (DE3) harboring pGEX-2T-*adsA* | *Thammavongsa et al., 2009* | N/A | |
| Cell line (*Homo sapiens*) | U937 | ATCC | ATCC CRL-1593.2 | |
| Cell line (*H. sapiens*) | U937 *DCK*[-/-] | *Winstel et al., 2018* | N/A | |
| Biological sample (human blood) | Blood samples from healthy donors | Hannover Medical School | N/A | |
| Antibody | α-CASP3, rabbit polyclonal | Cell Signaling | 9662 | 1:1000 |
| Antibody | α-GAPDH, rabbit monoclonal | Abcam | ab181602 | 1:10000 |
| Antibody | α-rabbit IgG, HRP-linked, goat polyclonal | Cell Signaling | 7074 | 1:10000 |
| Antibody | α-F4/80, rabbit monoclonal | Cell Signaling | 70076 | 1:150 |
| Antibody | α-rabbit IgG (H+L) Alexa Fluor 546, goat polyclonal | Invitrogen | A-11071 | 1:500 |
| Antibody | α-B220-PerCP/Cy5.5, rat monoclonal | BioLegend | 103236 | 1:100 |
| Antibody | α-CD4-BV711, rat monoclonal | BioLegend | 100550 | 1:100 |
| Antibody | α-CD8α-PE, rat monoclonal | BioLegend | 100708 | 1:100 |
| Antibody | α-CD11c-PE/Dazzle594, Armenian hamster monoclonal | BioLegend | 117347 | 1:100 |
| Antibody | α-CD11b-FITC, rat monoclonal | BioLegend | 101206 | 1:100 |
| Antibody | α-GR1-APC, rat monoclonal | BioLegend | 108412 | 1:100 |
| Antibody | α-CD16/32 (FC block), rat monoclonal | BioLegend | 101319 | 1:100 |
| Peptide, recombinant protein | Recombinant AdsA (rAdsA) | This study | N/A | |
| Commercial assay or kit | FITC Annexin V | BD | 556419 | |

*Continued on next page*

*Continued*

| Reagent type (species) or resource | Designation | Source or reference | Identifiers | Additional information |
|---|---|---|---|---|
| Commercial assay or kit | MojoSort Human CD14 Selection Kit | BioLegend | 480026 | |
| Commercial assay or kit | Caspase-3 Assay Kit | Sigma | CASP3C-1KT | |
| Chemical compound, drug | (*R*)-DI-87 | University of California, LA | N/A | |
| Chemical compound, drug | Captisol | CyDex Pharmaceuticals, Inc. | RC-0C7-020 | |
| Chemical compound, drug | Human macrophage colony stimulating factor | Genscript | Z02914 | |
| Chemical compound, drug | Mouse macrophage colony stimulating factor | Genscript | Z02930 | |
| Chemical compound, drug | Phorbol 12-myristate 13-acetate (PMA) | Sigma | P8139 | |

## Bacterial strains and growth conditions

All bacterial strains used in this study are listed in the Key Resources Table. Bacteria were grown in tryptic soy broth (TSB), Mueller-Hinton broth (MHB), or lysogeny broth (LB) at permissive temperatures. Media were supplemented with appropriate antibiotics (ampicillin 100 µg/ml).

## Cell lines and tissue culture

U937 cells were obtained from American Type Culture Collection (ATCC) and grown in Roswell Park Memorial Institute (RPMI) 1640 medium (Gibco) supplemented with 10% heat-inactivated fetal bovine serum (hi-FBS) according to the manufacturer's instructions. All mammalian cell lines were grown at 37 °C under 5% $CO_2$. Cell lines used in this study are listed in the Key Resources Table.

## Isolation of human primary cells

Peripheral blood mononuclear cells (PBMCs) were isolated from heparinized blood by density gradient centrifugation on Pancoll (PAN Biotech) according to standard laboratory protocols. Next, human primary monocytes were isolated from PBMCs by using magnetic nanobeads and the MojoSort Human CD14 Selection Kit (BioLegend) as described earlier (*Tantawy et al., 2022*). Subsequently, purified primary CD14-positive (CD14+) monocytes were resuspended in RPMI 1640 medium containing 10% hi-FBS and 1% penicillin-streptomycin and used for cytotoxicity assays. Alternatively, purified CD14+ monocytes were differentiated into human monocyte-derived macrophages (HMDMs) in the same medium supplemented with 50 ng/ml of human macrophage colony stimulating factor (hM-CSF; Genscript). HMDMs were used at day 7 post-differentiation for cytotoxicity experiments.

## Isolation of murine bone-marrow-derived macrophages

BMDMs were isolated from C57BL/6 mice as previously described (*Tantawy et al., 2022*). In brief, mice were euthanized to remove the femur and tibia which were sterilized by using 70% ethanol. Bones were then washed with sterile phosphate-buffered saline (PBS). Next, the bone ends were removed and the bone marrow was flushed out by using RPMI 1640 containing 10% hi-FBS and 1% penicillin-streptomycin. Following a resuspension step, cells were passed through a nylon cell strainer (40 µm) to remove unwanted tissue and cellular debris. Subsequently, cells were centrifuged (10 min, 200 x *g*, 4 °C), resuspended in red blood cell (RBC) lysis buffer (Roche), and incubated for 5 min at room temperature to lyse RBC. Cells were centrifuged once more, resuspended in BMDM medium (RPMI 1640 containing 10% hi-FBS, 1% penicillin-streptomycin, and 50 ng/ml of mouse macrophage colony-stimulating factor (Genscript)), and seeded into tissue culture-treated dishes to deplete bone marrow cells from fibroblasts. At day 1 post-extraction, suspension bone marrow cells were collected via centrifugation, adjusted to $6.0 \times 10^5$ cells/ml in BMDM medium, and re-seeded into bacteriological dishes. At day 4 post-extraction, cells were incubated with an additional 10 ml of BMDM medium. BMDMs were used at day 7 post-extraction for cytotoxicity experiments.

## Protein purification

Purification of recombinant *S. aureus* AdsA was performed as described elsewhere (*Thammavongsa et al., 2009*; *Tantawy et al., 2022*). Briefly, a glutathione S-transferase (GST)-tagged and recombinant

version of *S. aureus* AdsA (rAdsA) was expressed in *E. coli* BL21 using the pGEX-2T plasmid system (GE Healthcare) and purified via glutathione S-transferase affinity chromatography (*Thammavongsa et al., 2009*; *Tantawy et al., 2022*). Next, the N-terminal GST tag was removed by using thrombin which was immediately cleared from the protein sample via benzamidine sepharose beads according to the manufacturer's instructions (GE Healthcare). Purified rAdsA was analyzed by a Coomassie-stained SDS-PAGE following standard laboratory protocols.

## Cytotoxicity assays

To evaluate the protective role of (*R*)-DI-87 during death-effector deoxyribonucleoside-induced cytotoxicity, $2.0 \times 10^5$ U937 monocyte-like cells (U937) per well were seeded in a 24-well plate and pre-incubated for 2 hr at 37 °C under 5% $CO_2$ in RPMI 1640 medium containing 10% hi-FBS and 1 µM of (*R*)-DI-87. Control wells received vehicle only. Alternatively, $4.0 \times 10^5$ U937 cells per well were seeded in a 24-well plate and incubated for 48 hr at 37 °C under 5% $CO_2$ in RPMI 1640 growth medium that contained 160 nM phorbol 12-myristate 13-acetate (PMA). Resulting U937-derived macrophages (U937 MΦ) were washed and further incubated in RPMI 1640 growth medium lacking PMA (24 hr). Similarly, $3.5 \times 10^5$ BMDMs per well were seeded in 24-well plates and incubated for 24 hr at 37 °C under 5% $CO_2$ in corresponding growth media. Next, U937 MΦ or BMDMs were washed once and pre-incubated for 2 hr at 37 °C under 5% $CO_2$ in appropriate growth media supplemented with 1 µM of (*R*)-DI-87. Control wells received vehicle only. Following pre-incubation with (*R*)-DI-87, U937, U937 MΦ, or BMDMs were exposed to various concentrations of either dAdo or dGuo (100 µM for U937; 200 µM for U937 MΦ or BMDMs) and incubated for 48 hr at 37 °C under 5% $CO_2$ in RPMI 1640 growth medium. Cells were collected via centrifugation (U937) or a detachment-centrifugation step using trypsin-EDTA (U937 MΦ) or accutase (BMDMs) solution. Dead cells were stained with trypan blue and counted by using a microscope to calculate killing efficiency. To analyze the protective effect of (*R*)-DI-87 in primary human cells, $2.0 \times 10^5$ CD14$^+$ monocytes or HMDMs per well were seeded in multi-well plates and pre-incubated for 2 hr at 37 °C under 5% $CO_2$ in RPMI 1640 medium containing 10% hi-FBS and 1 µM of (*R*)-DI-87. Control wells received vehicle only. Subsequently, cells were intoxicated by using various concentrations of either dAdo or dGuo (100 µM for CD14$^+$ monocytes; 200 µM for HMDMs) and incubated in appropriate growth media for 48 hr at 37 °C under 5% $CO_2$. Cells were collected via centrifugation (CD14$^+$ monocytes) or a detachment-centrifugation step (HMDMs) using trypsin-EDTA solution. Viability of cells was determined via trypan blue staining and microscopy as described above. To block the cytotoxic effect of AdsA-derived death-effector deoxyribonucleosides by using (*R*)-DI-87, rAdsA (1.25 µg/µl) was incubated for 16 hr at 37 °C in a reaction buffer (30 mM Tris-HCl, pH 7.5; 1.5 mM $MgCl_2$; 1.5 mM $MnCl_2$) supplemented with either dAMP or dGMP (1.19 mM each) according to a published protocol (*Tantawy et al., 2022*). Controls included reactions that lacked deoxyribonucleoside monophosphates or rAdsA. Following incubation, all reaction products were filter-sterilized and added to U937 MΦ which were pre-incubated with vehicle or 1 µM (*R*)-DI-87 for 2 hr prior to intoxication. Next, cells were incubated at 37 °C under 5% $CO_2$ for 48 hr, collected, and analyzed via trypan blue staining and microscopy as described above. *S. aureus*-driven cell death was analyzed based on previously described approaches (*Thammavongsa et al., 2013*; *Winstel et al., 2018*; *Winstel et al., 2019*; *Tantawy et al., 2022*). In brief, the *S. aureus* Newman strain panel was incubated overnight at 37 °C in TSB, diluted in fresh TSB medium, and grown at 37 °C to $1.5 \times 10^8$ CFU/ml. Next, staphylococci were pelleted, washed twice in sterile wash buffer (50 mM Tris-HCl; pH 7.5), and adjusted to $3.2 \times 10^8$ CFU/ml. $8.0 \times 10^7$ CFU were incubated in a dAMP- or dGMP-containing (final conc. 5 mM) reaction buffer (30 mM Tris-HCl, pH 7.5; 2 mM $MgCl_2$) for 90 min at 37 °C. Controls lacked bacteria, dAMP, dGMP, or involved the *S. aureus adsA* mutant, which cannot synthesize dAdo and dGuo. Following incubation, bacteria were removed from the sample by a brief centrifugation-filtration step. 300 µl of the resulting and filter-sterilized supernatants were mixed with 700 µl of RPMI growth medium and incubated with vehicle- or (*R*)-DI-87-exposed (1 µM) U937-derived macrophages for 24 hr at 37 °C under 5% $CO_2$. Finally, cells were detached, collected, and stained with trypan blue as described above to quantify killed phagocytes.

## Assessment of caspase-3-activity

Caspase-3 activity was analyzed using a colorimetric caspase-3 detection kit (Sigma) and a published protocol (*Winstel et al., 2018*; *Tantawy et al., 2022*). Briefly, U937 cells or U937 MΦ along with

appropriate controls were pre-incubated for 2 hr at 37 °C under 5% $CO_2$ in RPMI 1640 medium containing 10% hi-FBS and 1 µM of (R)-DI-87. Controls received vehicle only. Subsequently, cells were exposed to various concentrations of either dAdo or dGuo (100 µM for U937; 200 µM for U937 MΦ) and incubated for 24 hr at 37 °C under 5% $CO_2$ in RPMI 1640 growth medium. Cells were collected via centrifugation (U937) or a detachment-centrifugation step (U937 MΦ) using trypsin-EDTA solution and washed once in PBS. Next, $1.0 \times 10^7$ cells were lysed in pre-chilled lysis buffer (Sigma kit) for 20 min. This step was performed on ice. Resulting lysates were centrifuged at 4 °C ($18,000 \times g$ for 10 min) to obtain cell- and debris-free supernatants which were incubated with the caspase-3 substrate Ac-DEVD-pNA according to the manufacturer's instructions. Caspase-3 activity was determined based on the amount of released pNA that can be detected at 405 nm.

## Immunoblotting

Immunoblotting was performed as described elsewhere (*Tantawy et al., 2022*). In short, U937 or U937 MΦ and appropriate controls were pre-incubated for 2 hr at 37 °C under 5% $CO_2$ in RPMI 1640 medium containing 10% hi-FBS and 1 µM of (R)-DI-87. Controls received vehicle only. Next, cells were treated with dAdo or dGuo (100 µM for U937; 200 µM for U937 MΦ), incubated for 24 hr at 37 °C under 5% $CO_2$ in RPMI 1640 growth medium, and collected via centrifugation (U937) or a detachment-centrifugation step (U937 MΦ) using trypsin-EDTA solution. Cells were washed once in PBS. $1.0 \times 10^7$ cells were lysed on ice in pre-chilled lysis buffer (50 mM HEPES, pH 7.4; 5 mM CHAPS; 5 mM DTT) for 20 min. Resulting lysates were centrifuged at 4 °C ($18,000 \times g$ for 10 min) to obtain cell- and debris-free supernatants which were mixed with sodium dodecyl sulfate-polyacrylamide gel (SDS-PAGE) loading buffer. Samples were boiled for 10 min at 95 °C. Proteins were separated via SDS-PAGE (12%) and transferred onto PVDF membranes for immunoblot analysis with the following rabbit primary antibodies: α-Caspase-3 (α-CASP3, 9662, Cell Signaling) and α-GAPDH (ab181602, Abcam, loading control). Immunoreactive signals were revealed with a secondary antibody conjugated to horseradish peroxidase (α-rabbit IgG, 7074, Cell Signaling). Horseradish peroxidase activity was detected with enhanced chemiluminescent (ECL) substrate (Thermo Fisher).

## FITC-annexin-V/PI staining

FITC-annexin-V/PI staining of U937 MΦ or HMDMs exposed to 1 µM (R)-DI-87 and death-effector deoxyribonucleosides (200 µM dAdo or dGuo; 24 hr at 37 °C) along with appropriate controls was performed by using FITC-annexin-V Apoptosis Detection Kit I (BD Biosciences) according to the manufacturer's instructions. Stained cells were analyzed via immunofluorescence microscopy according to standard laboratory protocols.

## Determination of the minimal inhibitory concentration (MIC)

The MIC of (R)-DI-87 was determined in 96-well plates by using the microdilution method and Mueller-Hinton broth according to standard laboratory protocols. Wells containing varying concentrations of (R)-DI-87 or vehicle were inoculated with $10^5$ CFU/ml of wild-type *S. aureus* Newman or its *adsA* mutant and incubated for 24 hr at 37 °C under continuous shaking. The MIC was defined as the lowest concentration of compound at which no visible growth was detected.

## Bacterial survival in blood

Bacterial survival in mouse or human blood was analyzed as described before (*Thammavongsa et al., 2009*). In brief, fresh overnight cultures of *S. aureus* Newman wild type or its *adsA* mutant were diluted into fresh TSB medium and grown at 37 °C to an OD of 1.0. Bacteria were washed twice in sterile PBS and adjusted in PBS to a final density of $1.0 \times 10^8$ CFU/ml. Next, freshly drawn mouse or human blood anticoagulated with heparin was incubated in the presence or absence of 1 µM (R)-DI-87 for 60 min (37 °C) with *S. aureus* Newman wild type or its *adsA* mutant using a bacterial dose of $1.0 \times 10^6$ CFU/ml (murine blood) or $1.0 \times 10^7$ CFU/ml (human blood). Samples without (R)-DI-87 treatment received the vehicle. Following incubation, blood samples were mixed in a 1:1 ratio with sterile lysis buffer (PBS containing 1.0% saponin) and incubated for 10 min at 37 °C to lyse eukaryotic cells. Subsequently, serial dilutions were prepared and plated onto TSA plates to determine bacterial survival rates.

## Animal infection model

C57BL/6 mice were purchased from Janvier Laboratories and kept under specific pathogen-free conditions in our central mouse facility (TWINCORE, Center for Experimental and Clinical Infection

Research, Hannover, Germany). Mice received (*R*)-DI-87 (75 mg/kg) or vehicle (40% Captisol) via oral gavage at the day of the infection, separated by 12 hr intervals over a 5-day observation period. Timing of intervention therapy was initiated 6 hr before systemic challenge. For infection experiments, TSB overnight cultures of wild-type *S. aureus* Newman or its *adsA* mutant were diluted 1:100 in TSB and grown to an optical density (600 nm) of 0.5. Bacteria were then centrifuged (10 min, RT, 8000 × *g*), washed twice in sterile PBS, and adjusted to $10^8$ CFU/ml. One hundred microliters of the bacterial suspension ($10^7$ CFU) were administered intravenously (lateral tail vein) into 6- to 8-weeks-old female C57BL/6 mice. Five days post-infection, animals were euthanized. Organs were dissected, examined for surface abscesses, and homogenized in sterile PBS supplemented with 0.1% Triton X-100. Serial dilutions were prepared and plated onto TSA plates to determine bacterial loads. For histopathology, dissected organs were fixed in 10% Formalin (Sigma), embedded into paraffin, thin-sectioned, and stained with hematoxylin and eosin (H&E). Stained tissues were examined by microscopy according to standard laboratory protocols.

## Immunofluorescence staining

To detect macrophages in *S. aureus*-infected tissues, formalin-fixed and paraffin-embedded organs were thin-sectioned, deparaffinized, and rehydrated. After heat-induced antigen retrieval in 10 mM sodium citrate buffer (pH 6.0), non-specific antibody binding was blocked by adding 2% normal goat serum. Immunofluorescence staining was carried out by using an antibody against F4/80-positive macrophages (α-F4/80, 70076, Cell Signaling), followed by a fluorescently labeled secondary antibody (Alexa546, Invitrogen). Stained tissues were examined by using a Zeiss Apotome 2 microscope (Zeiss).

## Safety assessment of (*R*)-DI-87

To assess the therapeutic value and safety of (*R*)-DI-87, cohorts of female C57BL/6 mice were treated with (*R*)-DI-87 (75 mg/kg) or vehicle (40% Captisol) via oral gavage in 12 hr intervals over the course of 23 days. On day 16, 100 µl peripheral blood was collected in lithium heparin-coated tubes via retro-orbital bleeding using heparin-coated capillary tubes. Subsequently, blood samples were incubated with 5 ml of ACK lysis buffer at room temperature for 5 min, quenched with 5 ml of FACS buffer (5% FBS in PBS), and centrifuged at 4 °C (4 min). This process was repeated and cells were subsequently stained with the following fluorochrome-conjugated anti-mouse antibodies diluted 1:100 in 100 µl of FACS buffer for 20 min at 4 °C: α-B220-PerCP/Cy5.5 (103236, BioLegend), α-CD4-BV711 (100550, BioLegend), α-CD8α-PE (100708, BioLegend), α-CD11c-PE/Dazzle594 (117347, BioLegend), α-CD11b-FITC (101206, BioLegend), α-GR1-APC (108412, BioLegend); α-CD16/32 (FC block; 101319; BioLegend). Following incubation, cells were centrifuged and washed twice using FACS buffer. Cells were resuspended in FACS buffer and analyzed using a BD LSRII flow cytometer and the FlowJo software package. During the (*R*)-DI-87 or vehicle treatment procedure, mice were also regularly monitored and weighed to assess the overall health status. On day 23, an endpoint analysis was carried out to analyze the immune cell profile and cellularity of spleen and thymus tissues as described elsewhere (*Abt et al., 2022*). Lastly, blood was collected from all cohorts of animals for the analysis of the deoxycytidine (dC) content in mouse plasma to evaluate the dCK-inhibitory capacity of (*R*)-DI-87 (see below).

## LC-MS/MS-MRM analysis of plasma nucleoside levels

LC-MS/MS-MRM analysis of plasma nucleoside levels was performed as previously described (*Le et al., 2017*). For the analysis of plasma deoxycytidine levels, blood was collected from mice using a heparin-coated capillary tube by the retro-orbital technique and transferred to a lithium-heparin coated tube (Fisher Cat#13-680-62) on ice. Samples were centrifuged at 450 x *g* for 5 min at 4 °C and the plasma supernatant was stored at –80 °C. For metabolite extraction, 20 µL of plasma was mixed with 80 µL of 100% MeOH containing stable isotope-labeled nucleoside internal standard (0.5 µM [U-$^{15}$N/$^{13}$C]dC; Silantes Cat#124603802). MeOH-extracted samples were incubated at –80 °C for 24 hr, centrifuged at 12,000 x *g* for 5 min at 4 °C, and the cleared supernatant was transferred to an HPLC injector vial for analysis. Five µL of the sample was injected onto a porous graphitic carbon column (Thermo Fisher Scientific Hypercarb, 100x2.1 mm, 5 µm particle size) equilibrated in solvent A (0.1% formic acid in MiliQ-purified/LC-Pak treated $H_2O$) and eluted (700 µL/min) with an increasing concentration of solvent B (0.1% formic acid in acetonitrile) using min/%B/flow rates (µL/min) as follows:

0/2/700, 3/80/700, 4/80/700, 4.5/2/700, 7/2/700. The effluent from the column was directed to an Agilent Jet Stream ion source connected to a triple quadrupole mass spectrometer (Agilent 6460) operating in the multiple reaction monitoring (MRM) mode using previously optimized settings. The peak areas for each target molecule (precursor→fragment ion transitions) at predetermined retention times were recorded using Agilent MassHunter software. Peak areas were normalized to nucleoside internal standard signals. An external standard curve was applied to determine nucleoside concentrations. Experimental and standard samples were processed together to minimize variation.

## Synthesis and preparation of (*R*)-DI-87

(*R*)DI-87((((*R*)–2-((1-(2-(4-methoxy-3-(2-morpholinoethoxy)phenyl)–5-methylthiazol-4-yl)ethyl)thio)pyrimidine-4,6-diamine))) was synthesized as described before (*Poddar et al., 2020*). For experiments, (*R*)-DI-87 was either dissolved in Dimethyl sulfoxide (*in vitro* experiments) or in an aqueous solution containing 40% Captisol (CyDex Pharmaceuticals, Inc) (*in vivo* experiments).

## Statistical analysis

Statistical analysis was performed by using GraphPad Prism (GraphPad Software, Inc, La Jolla, USA). Statistically significant differences were calculated by using statistical methods as indicated. p-values <0.05 were considered significant.

## Acknowledgements

We thank all laboratory members for helpful discussion. We are grateful for support from the German Research Foundation (award WI4582/2-1 to VW, project number 449712894). ERA is supported by UCLA Tumor Immunology Training Grant T32CA009120. This work was supported by National Institutes of Health grants to CGR (1R01CA25052901A1 and 1R01CA260678-01).

# Additional information

### Competing interests

Volker Winstel, Evan R Abt: Inventor in a patent application covering the use of dCK inhibitors as a treatment for bacterial infectious diseases (patent pending, No. 63/450,304 USPTO). Caius G Radu: Co-inventor of the dCK inhibitor used in this study. This intellectual property has been patented by the University of California and optioned to Trethera Corporation, a company that CGR owns equity in (US 9,598,404; US 9,688,673). Inventor in a patent application covering the use of dCK inhibitors as a treatment for bacterial infectious diseases (patent pending, No. 63/450,304 USPTO). The other author declares that no competing interests exist.

### Funding

| Funder | Grant reference number | Author |
| --- | --- | --- |
| Deutsche Forschungsgemeinschaft | WI4582/2-1 | Volker Winstel |
| University of California, Los Angeles | T32CA009120 | Evan R Abt |
| National Institutes of Health | 1R01CA25052901A1 | Caius G Radu |
| National Institutes of Health | 1R01CA260678-01 | Caius G Radu |

The funders had no role in study design, data collection and interpretation, or the decision to submit the work for publication.

### Author contributions

Volker Winstel, Conceptualization, Data curation, Formal analysis, Supervision, Funding acquisition, Investigation, Visualization, Methodology, Writing – original draft, Project administration, Writing

– review and editing; Evan R Abt, Thuc M Le, Conceptualization, Data curation, Formal analysis, Investigation, Methodology, Writing – review and editing; Caius G Radu, Conceptualization, Resources, Formal analysis, Funding acquisition, Investigation, Methodology, Writing – review and editing

**Author ORCIDs**
Volker Winstel 🔟 https://orcid.org/0000-0002-3248-2572

**Ethics**

Human subjects: Human blood samples were obtained from adult, consenting healthy donors. Informed consent forms were obtained from all participants. These studies were reviewed and approved by the medical ethics committee of Hannover Medical School (Hannover, Germany) under the permission number 8831_BO_K_2019.

All animal experiments were conducted in accordance with the local animal welfare regulations reviewed by the institutional review board and the Niedersächsisches Landesamt für Verbraucherschutz und Lebensmittelsicherheit (LAVES) under the permission number 33.19-42502-04-21/3696. Scientific use of animals (murine blood samples; isolation of BMDMs) was further carried out under approved animal care and use protocols which are granted by the state veterinary authorities and overseen by the internal animal care and use committee (IACUC). Animal studies required to assess the safety of (R)-DI-87 in mice were approved by the UCLA Animal Research Committee (ARC).

Reviewer #1 (Public Review): https://doi.org/10.7554/eLife.91157.3.sa1
Author Response https://doi.org/10.7554/eLife.91157.3.sa2

---

# Additional files

**Supplementary files**
• Supplementary file 1. Minimum inhibitory concentration of (R)-DI-87.
• MDAR checklist

**Data availability**

All data supporting the findings of this study are available within the article, its supplementary material, and its source data files.

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
