## [Editor Report · eLife assessment]

This **valuable** study combines *in vitro* and *in vivo* experiments designed to test if a deoxycytidine kinase inhibitor provides therapeutic benefit during infection with *Staphylococcus aureus*. The authors provide **compelling** evidence that this putative host-directed therapy has good potential to promote natural clearance of infection without targeting the bacterium. This paper would be of interest to bacteriologists, immunologists, and those studying host-microbe interactions.

---

## [Referee Report · Reviewer #1 (Public Review)]

Aiming at the problem that *Staphylococcus aureus* can cause apoptosis of macrophages, the authors found and verified that drug (R)-DI-87 can inhibit mammalian deoxycytidine kinase (dCK), weaken the killing effect of *Staphylococcus aureus* on macrophages, and reduce the apoptosis of macrophages. And increase the infiltration of macrophages to the abscess, thus weakening the damage of *Staphylococcus aureus* to the host. This work provides new insights and ideas for understanding the effects of *Staphylococcus aureus* infection on host immunity and discovering corresponding therapeutic interventions. This work is important and groundbreaking.

Comments on revised version:

The changes made by the authors addressed my previous concerns about the manuscript and greatly improved the quality of the article.

---

## [Author Response]

The following is the authors’ response to the original reviews.

We thank the editor for organizing the review of our manuscript. We have carefully read and analyzed the reviewers’ comments, addressed each criticism point-by-point as outlined below, and modified the manuscript and figures accordingly. In this regard, we would also like to take the opportunity to thank both reviewers for their thoughtful suggestions for improvement of our manuscript. We believe that our manuscript has improved as a result, and hope that it is now suitable for publication.

**Public Reviews:**

**Reviewer #1 (Public Review):**
Aiming at the problem that *Staphylococcus aureus* can cause apoptosis of macrophages, the author found and verified that drug (R)-DI-87 can inhibit mammalian deoxycytidine kinase (dCK), weaken the killing effect of *Staphylococcus aureus* on macrophages, and reduce the apoptosis of macrophages. And increase the infiltration of macrophages to the abscess, thus weakening the damage of *Staphylococcus aureus* to the host. This work provides new insights and ideas for understanding the effects of *Staphylococcus aureus* infection on host immunity and discovering corresponding therapeutic interventions.The logic of the study is commendable, and the design is reasonable.Some data related to the conclusion of the paper need to be supplemented, and some experimental details need to be described.

Response: We thank the reviewer for the positive feedback along with the detailed and knowledgeable analysis of this paper. Specific details and comments on all raised concerns can be found below.

**Reviewer #2 (Public Review):**
Summary:In this study, Winstel and colleagues test if the deoxycytidine kinase inhibitor, (R)-DI-87 provides therapeutic benefit during infection with *Staphylococcus aureus*. The premise behind the current work is a series of prior studies that found that *S. aureus* can disable functional immune clearance by generating NET-derived deoxyribonucleosides to induce macrophage apoptosis via purine salvage. Here, the authors use in vitro and in vivo experiments with (R)-DI-87 to demonstrate that inhibition of deoxycytidine kinase prevents *S. aureus*-induced deoxyribonucleoside-mediated macrophage cell death, to bolster immune cell function and promote more effective clearance during infection. The authors conclude that (R)-DI-87 represents and potentially important Host-Directed Therapy (HDT) with good potential to promote natural clearance of infection without targeting the bacterium. Overall, the study represents an important next step in the exploration of purine salvage and deoxyribonucleoside toxicity as a targetable pathway to bolster infection clearance and provides early-stage evidence of the therapeutic potential of (R)-DI-87 during *S. aureus* infection.

Response: We thank the reviewer for the thoughtful suggestions for improvement of our manuscript.Specific details and comments on all raised concerns can be found below.

Strengths:The study has several strengths that support its conclusions:(1) Well-controlled in vitro studies that firmly establish (R)-DI-87 is capable of blocking deoxyribonucleoside-mediated apoptosis of immune cell lines and primary cells.(2) Solid evidence to support that administration of (R)-DI-87 can have therapeutic benefits during infection (reduced number of abscesses and reduced CFU).(3) Controls included to ascertain the degree to which (R)-DI-87 might have secondary effects on immune cell distribution.(4) Controls included to ascertain whether or not (R)-DI-87 has intrinsic antibacterial properties.Weaknesses:However, there are several important weaknesses related to the rigor of the research and the conclusions drawn. The most relevant weaknesses noted by this reviewer are:(1) Drawing firm conclusions about the therapeutic potential of (R)-DI-87 using only *S. aureus* strain Newman, a methicillin-susceptible *S. aureus*, that while a clinical isolate is not clearly representative of the strains of S. aureus causing infection in hospitals and communities. Newman also harbors an unusual mutation in a regulator that dramatically changes virulence factor gene expression. While the data with Newman remains valuable, the absence of consideration of other strains, including MRSA, makes it more difficult to support the relatively broad conclusions about therapeutic potential made by the authors.

Response: We assume that this is a misunderstanding. *S. aureus* Newman is a patient-derived isolate and not a regulator mutant and/or laboratory strain (Duthie and Lorenz LL 1952, J Gen Microbiol 6(1-2), 95107). Its genome is fully sequenced (Baba et al. 2008, J Bacteriol 190(1):300-10) and it is highly virulent in mouse or human ex vivo models (e.g. Alonzo 3rd et al. 2013, Nature 493(7430):51-5.; DuMont et al. 2011, Mol Microbiol 79(3):814-25; Skaar et al. 2004, Science 305(5690):1626-8). Moreover, *S. aureus* Newman has served as a gold standard to study abscess formation in the past (e.g. Thammavongsa et al. 2013, Science 342(6160):863-6; Cheng et al. 2009, FASEB J 23(10):3393-404; Corbin et al. 2008, Science 319(5865):962-5) and has further also been used multiple times to test the therapeutic efficacy of antimicrobial or anti-infective agents in various animal models of infectious disease (e.g. Buckley et al. 2023, Cell Host Microbe 31(5):751-765.e11; Zhang et al. 2014, PNAS 111(37):13517-22; Richter et al. 2013, PNAS 110(9):3531-6). Apart from this, it is crucial to note that methicillin-sensitive isolates such as *S. aureus* Newman are typically more frequently isolated in hospitals as compared to MRSA. Specifically, public health system- and population-based surveillance studies clearly indicate that annual incidence rates for MSSA infections are dominant over those associated with MRSA infections (e.g. Gagliotti et al. 2021, Euro Surveill 26(46):2002094; Jackson et al. 2020, Clin Infect Dis 70(6):1021-1028; Laupland et al. 2013, Clin Microbiol Infect 19(5):465-71), even in groups at elevated risk (e.g. McMullan et al. 2016, JAMA Pediatr et al., 170(10):979-986; Ericson et al. 2015, JAMA Pediatr 169(12):1105-11). Although we understand and agree with the reviewer that certain MRSA clones can be a dominant cause of staphylococcal disease in specific geographic areas, we believe that *S. aureus* Newman adequately reflects staphylococcal isolates that cause the majority of infections in humans. In this regard, we would also like to highlight once more that (R)-DI-87 targets host dCK and not the bacterium. Accordingly, the antibiotic resistance status of *S. aureus* is not expected to impact our main findings and conclusions as (R)-DI-87 exclusively inhibits dCK, a key element of the mammalian purine salvage pathway.

(2) In vitro (R)-DI-87 efficacy studies with dAdo and dGuo are strong, however, the authors do not test the in vitro efficacy of (R)-DI-87 using *S. aureus*. They have done this type of work in prior studies (See doi: 10.1073/pnas.1805622115 - Figure 5). If included it would greatly strengthen their argument that (R)-DI87 is directly affecting the *S. aureus*  Nuclease  AdsA macrophage-killing pathway. Without it, the evidence provided remains indirect, and several conclusions may be overstated.

Response: We highly appreciate this comment and agree with the reviewer that such an experiment would support our main findings. Thus, we have performed additional experiments and took advantage of a previously described approach (Tantawy et al. 2022, Front Immunol 13:847171) to demonstrate that (R)DI-87-mediated inhibition of host dCK enhances macrophage survival upon treatment with culture media that had been conditioned by incubation with adsA-proficient or adsA-deficient staphylococci in the presence or absence of purine deoxyribonucleoside monophosphates. Our findings are described in the main text and in a new figure (Fig. 2K-L). Based on these new findings and together with our rAdsA-based approach (Fig. 2I-J), we are confident that (R)-DI-87 represents a suitable small molecule inhibitor of host dCK which can prevent host immune cell death induced by toxigenic products associated with the *S. aureus* Nuc/AdsA pathway.

(3) Caspase-3 immunoblot experiments seem to suggest an alternative conclusion to what was made by the authors. They point out that Caspase-3 cleavage does not occur upon treatment with (R)-DI-87. However, the data seem to argue that there is almost no caspase-3 present in (R)-DI-87 treated cells (cleaved or uncleaved). Might this suggest that caspase-3 is not even produced when cells are not experiencing deoxyribonucleoside toxicity? Perhaps the authors could reconsider the interpretation of this data.

Response: We believe that this is a misunderstanding. Our immunoblots (Fig. 3E-F) show only the processed forms of caspase-3. The antibody we have used can recognize full-length caspase-3 along with the p17 and p19 subunits that can result from cleavage. To clarify this point, we have slightly modified our main figure and provide the full immunoblots (Source data file) which clearly demonstrate that unprocessed caspase-3 (pro-caspase-3) is present in all samples. In this regard, we further note that caspase-3 can also form heterocomplexes with other proteins, presumably explaining some of the unknown bands in samples obtained from cells that have been exposed to death-effector deoxyribonucleosides. Additional bands are probably a result of cross-reactivity of the antibody and/or unspecific degradation of pro-caspase in cellular lysates.

(4) There are some concerns over experimental rigor and clarity of the experimental design in the methods. The most important points noted by this reviewer are included here. (a.) There is no description of the number of replicates or representation of the Western blots and no uncropped blots are provided. (b.) the methods describing the treatment conditions for in vivo studies are not sufficiently clear. For example, it is hard to tell when (R)-DI-87 is first administered to mice. Is it immediately before the infection, immediately after, or at the same time? This has important implications for interpreting the results in terms of therapeutic potential. (c.) There are several statements made that (R)-DI-87 does not have a negative impact on the mice however, it is not sufficiently clear that the studies conducted are sufficient to make this broader claim that (R)-DI-87 has no impact on the animal, except as it relates to the distribution of immune cells, which is directly tested. (d.) there are no quantitative measures of apoptosis or macrophage infiltration, which impacts the rigor of these imaging experiments. (e.) only female mice are used in the in vivo studies. There is no justification provided for this choice; however, the rigor of the study design and the ability to draw conclusions about therapeutic potential is impacted in the absence of consideration of both sexes.

Response: Thank you for raising these points here. (a) We have modified our figure legend and provide the full immunoblots (Source data file) in order to clarify this point. (b) Moreover, we now provide more experimental details on the treatment conditions that were used to administer (R)-DI-87 to mice (methods section). (c) Furthermore, we have conducted new experiments in order to demonstrate that administration of (R)-DI-87 has no impact on laboratory animals. Specifically, we provide new data along with additional text on organ cellularity following long-term exposure of mice to (R)-DI-87. In this regard, we have also applied our immuno-phenotyping approach to spleen tissues samples derived from mice that received (R)-DI-87 or vehicle. As outlined in our new results, neither developmental errors nor differences in lymphocyte development have been observed (new Fig. 4B-C; new supplementary Fig. 3). Together with our data on mouse body weight along with our immuno-phenotyping approach of blood cells (Fig. 4A and 4D) and the fact that (R)-DI-87 is extremely well tolerated in humans (personal communication; KennethA. Schultz, Trethera Corporation, Los Angeles, CA, USA), we are very confident that application of (R)-DI87 is safe and has no detrimental impact on the host. (d) Lastly, we would like to point out that due to the densely packed and extremely sticky cuff of immune cells within staphylococcal abscesses, it is technically not possible to extract enough abscess material required for a reliable quantification of apoptotic macrophages within infectious foci. Such an analysis would also not allow us to differentiate between lesion-infiltrating macrophages and macrophages that may reside at the periphery of the abscess. For these reasons, we have established a fluorescence microscopy-based approach to demonstrate increased macrophage infiltration rates into abscesses formed in organs of mice that have been treated with the dCK-specific inhibitor (R)-DI-87 (Fig. 5A-P). Nonetheless, we have slightly modified our figure and its legend in order to help the readership to localize *S. aureus*-derived tissue lesions and the periphery of abscesses in these images. (e) Finally, publicly available databases indicate that dCK is equally well expressed in various tissues in both sexes. Moreover, dCK is not encoded on a sex chromosome, neither in mice nor in humans. Thus, we believe that it is justified to test the in vivo efficacy of (R)-DI-87 in female mice. Nonetheless, we have conducted additional in vitro experiments to test whether (R)-DI-87 can protect male animal-derived BMDMs from death-effector deoxyribonucleosides in a manner similar to cells derived from female mice. As expected, we did not observe a sex-specific effect (new supplementary Fig. 5), and hope that this adequately addresses this point.

(5) Animal studies show significant disease burden (CFU) even after administration of (R)-DI-87. Given the absence of robust clearance of infection, the author's claims read as an overstatement of the data. The authors may wish to reframe their conclusions to better highlight the potential benefit of this therapy at reducing severe disease but also to point out relevant limitations, especially considering that it does not lead to clearance in this model. In general, the consideration of the limitations of the proposed therapeutic approach, as uncovered by the data, is not present. A more nuanced consideration of the data and its interpretations, including both strengths and limitations, would greatly help to frame the study.

Response: Thank you for raising this point here. To highlighting the limitations of our approach, we have modified several passages in the main text. Moreover, we have adjusted our discussion section accordingly.

**Reviewer #1 (Recommendations For The Authors):**
(1) In vivo experiments, the dose given to mice was 75mg/kg. How did the author determine the dose of this drug?

Response: We thank the reviewer for this question, which gives us the chance to clarify this point. The experimental condition used to block host dCK in mice has been adopted from a previous publication (Chen et al. 2023, Immunology 168(1):152-169). To improve the overall quality of our current manuscript, we now included more background information addressing this point. Specifically, we have added additional in vivo and biochemical data along with more conclusive text to our results section to better explain the reason for the dose given to mice (new Fig. 4E).

(2) The author established a mouse model of *Staphylococcus aureus* blood infection in vivo and divided four groups for related experiments. It is suggested that the authors should supplement the survival rate of mice in each group so that readers can understand the effect of the drug on the survival of mice with bloodstream infection.

Response: While this is an interesting suggestion by the reviewer, we believe that this is beyond the scope of our study. In particular, the current study focused on analyzing the capacity of the dCK-specific inhibitor (R)-DI-87 to improve macrophage survival during staphylococcal abscess formation in an effort to lower bacterial loads in infected organ tissues. However, we agree with the reviewer that (R)-DI-87 might also help to improve further clinical syndromes of staphylococcal infections, including lethal bloodstream infection. We therefore modified parts of our discussion to address this point.

(3) In the in vivo experiment, the author administered the drug by intragastric administration, but the treatment was for the bloodstream infection of *Staphylococcus aureus*, so the author needed to determine the actual effective concentration of the drug in the blood of mice.

Response: We thank the reviewer for this comment and agree that inclusion of more background information and data would be a valuable addition to our manuscript. As outlined above, we have designed our in vivo experiments based on the methodology of a previous publication (Chen et al. 2023, Immunology 168(1):152-169). Similar to Chen and colleagues, we have also used a dose of 75 mg/kg of (R)-DI-87 that allows complete inhibition of host dCK in vivo. In this regard, we have now performed additional in vivo experiments to address this point. More precisely, we took advantage of a highly sensitive and LC-MS/MSbased method to measure accumulation of deoxycytidine, the natural substrate of host dCK, in mouse plasma upon administration of the dCK-specific inhibitor. As shown in our new Fig. 4E, administration of (R)-DI-87 at a dose of 75 mg/kg strongly increased deoxycytidine levels in mouse plasma thereby indicating that host dCK activity is completely blocked under these experimental conditions.

(5) This work is to reduce the apoptosis of macrophages through drug inhibition of dck, but not directly inhibit the related virulence of *Staphylococcus aureus*. Therefore, it is suggested that the author modify the title to summarize the whole paper more accurately.

Response: We agree with the reviewer that our manuscript’s title might be a bit misleading as (R)-DI-87 does not directly target the bacterium or staphylococcal virulence factors. Thus, we have modified the title of our revised manuscript to: “Targeting host deoxycytidine kinase mitigates *Staphylococcus aureus* abscess formation”.